# Obesity- and High-Fat-Diet-Induced Neuroinflammation: Implications for Autonomic Nervous System Dysfunction and Endothelial Disorders

**DOI:** 10.3390/ijms26094047

**Published:** 2025-04-25

**Authors:** Galateia Stathori, Nikolaos F. Vlahos, Evangelia Charmandari, Georgios Valsamakis

**Affiliations:** 1Center for the Prevention and Management of Overweight and Obesity, Division of Endocrinology, Metabolism and Diabetes, First Department of Pediatrics, Medical School, National and Kapodistrian University of Athens, ‘Aghia Sophia’ Children’s Hospital, 11527 Athens, Greece; g.stathor@gmail.com (G.S.); evangeliacharmandari@googlemail.com (E.C.); 2Second Department of Obstetrics and Gynecology, Medical School, National and Kapodistrian University of Athens, ‘Aretaieion’ University Hospital, 11528 Athens, Greece; gynoffice04@gmail.com; 3Division of Endocrinology and Metabolism, Center of Clinical, Experimental Surgery and Translational Research, Biomedical Research Foundation of the Academy of Athens, 11527 Athens, Greece

**Keywords:** obesity, high-fat diets, hypothalamic inflammation, neuroinflammation, autonomic nervous system dysfunction, endothelial disorders

## Abstract

Obesity is a multifactorial condition linked to severe health complications, including cardiovascular diseases and endothelial dysfunction. Both obesity and high-fat diets (HFDs) are strongly associated with neuroinflammation, particularly in the hypothalamus. The autonomic nervous system (ANS), which controls involuntary physiological processes, is critical for maintaining cardiovascular health, and its dysfunction is implicated in endothelial disorders. With its homeostatic control centers located in the hypothalamus and brainstem, a crucial question arises: could obesity- and HFD-induced neuroinflammation disrupt central ANS structures, leading to ANS dysfunction and subsequent endothelial disorders? This review examined whether neuroinflammation caused by obesity and HFD contributes to endothelial dysfunction through the dysregulation of the ANS. Our analysis revealed that hypothalamic inflammation linked to obesity and an HFD is associated with sympathetic hyperactivity and endothelial disorders. Identified molecular mechanisms include the influence of inflammatory cytokines, activation of the NF-κB/IKK-β pathway, microglial activation mediated by angiotensin II, circulating mitochondria triggering cGAS activation, and the stimulation of the TLR4 pathway. Our findings suggest that hypothalamic inflammation may play a central role in the interplay between obesity/an HFD, ANS dysfunction, and endothelial disorders.

## 1. Introduction

Obesity is a complex, multifactorial condition marked by the excessive accumulation of adipose tissue, leading to significant health issues [1]. Its pathophysiology involves a complex interaction between genetic predisposition, environmental factors, and metabolic dysregulation [1]. Key mechanisms include a chronic energy imbalance, altered lipid metabolism, and the dysregulation of appetite and satiety hormones [1]. Obesity is associated with numerous complications, including insulin resistance, type 2 diabetes mellitus, cardiovascular diseases, certain cancers, and musculoskeletal disorders [1]. A high-fat diet (HFD) is widely recognized as a major contributor to the onset of obesity, which is now considered a chronic low-grade inflammatory state due to the secretion of pro-inflammatory molecules from adipose tissue [2]. Recently, obesity and HFDs have been linked to central hypothalamic inflammation in humans, with increased neural gliosis [3]. The autonomic nervous system (ANS) is a crucial part of the nervous system responsible for regulating involuntary physiological processes, such as the heart rate, blood pressure, digestion, and respiratory rate [4]. It is divided into the sympathetic and parasympathetic nervous systems, which have opposing effects on target organs to maintain homeostasis [4]. The efferent activity of the ANS is mainly controlled by autonomic reflexes, with sensory information sent to homeostatic control centers in the hypothalamus and brainstem [4]. The vagus nerve (cranial nerve X) and other cranial nerves transmit sensory input from the thoracic and abdominal viscera to these centers [4]. This input is integrated, and nerve signals are sent to adjust the activity of preganglionic autonomic neurons, ensuring optimal functioning of the body’s internal environment [4].

Additionally, the ANS interacts with the endothelium through both sympathetic and parasympathetic innervation of blood vessel walls, regulating contraction and wall tension [5]. Sympathetic nerve fibers innervate the muscular layer of vessel walls, while cholinergic nerve endings are present in both the muscular and endothelial layers [5]. ANS imbalances can lead to cardiovascular disorders through mechanisms such as increased blood pressure, vasoconstriction, and reduced venous capacitance [6]. Cardiac autonomic dysfunction with decreased heart rate variability has been associated with type 2 diabetes, myocardial infarction, and sudden death [6]. Thus, the ANS plays a pivotal role in cardiovascular health, with its dysfunction contributing to various cardiovascular diseases.

Obesity, in addition to its association with cardiovascular risk factors, has also been linked to endothelial dysfunction [7]. The exact mechanisms underlying this relationship are not fully understood. However, obesity and HFDs have been associated with neuroinflammation, particularly of the hypothalamus [3]. Given that the central regulators of the ANS are located within the CNS, especially the hypothalamus, obesity-mediated hypothalamic inflammation could similarly affect ANS neurons. This raises the question of whether hypothalamic inflammation-mediated dysregulation of the ANS could lead to endothelium disorders.

In this review, we aimed to explore whether obesity and HFD-mediated neuroinflammation could be associated with endothelium disorders through ANS dysfunction.

## 2. The Role of Obesity and an HFD in Hypothalamic Inflammation

Obesity is recognized as a chronic disease characterized by low-grade inflammation. Visceral fat, in particular, has been associated with systemic inflammation due to its direct access to the portal circulation, through which it releases inflammatory molecules [2] (p. 2). The adipose tissue of obese individuals has been found to secrete pro-inflammatory cytokines, such as TNF-a, IL-1, and IL-6. In contrast, the adipose tissue of lean individuals predominantly secretes anti-inflammatory cytokines, like IL-4 and IL-10 [2] (p. 2). Additionally, immune B-cells in obese individuals produce cytokines that act locally on adipose tissue, thereby triggering inflammation [2] (p. 2). Moreover, obese individuals tend to have less brown adipose tissue and higher amounts of white adipose tissue, which is implicated in inflammation and immunity [8].

Besides systemic low-grade inflammation, obesity has been linked to central neuroinflammation, particularly in the hypothalamus. Specifically, in humans, obesity has been associated with inflammation of the left mediobasal hypothalamus, which is significantly correlated with low-grade peripheral inflammation [3] (p. 2). Interestingly, both in animals and humans, a high-fat diet (HFD) has also been linked to hypothalamic inflammation, either in conjunction with or independent of obesity [3,9].

There are two primary proposed mechanisms for the connection between obesity/an HFD and neuroinflammation. The first mechanism involves obesity-related blood–brain barrier dysfunction, which permits the entry of inflammatory cytokines from the periphery into central nervous system structures [10]. These cytokines then activate local microglia and astrocytes, which, in turn, secrete additional pro-inflammatory cytokines, exacerbating neuroinflammation. The second proposed mechanism involves the direct activation of microglia by HFD through increased fatty acids. This activation relies on the Toll-like receptor (TLR)–nuclear factor-κB (NF-κB) pathway, which promotes the expression of pro-inflammatory cytokines [10].

It is also worth noting that scientific evidence suggests that altered gut microbiota may play a role in the interplay between HFD and neuroinflammation. However, the exact mechanism behind this involvement requires further investigation [11].

## 3. Obesity, High-Fat Diet, and Vascular Endothelial Dysfunction

The endothelium is a thin layer of cells lining the interior surface of blood vessels, acting as a critical barrier and interface between the bloodstream and the vascular wall [12]. As a paracrine organ, the endothelium secretes various signaling molecules, such as nitric oxide, prostacyclin, and endothelin, which regulate the vascular tone, blood flow, and hemostasis. Additionally, it plays a pivotal role in immune responses and inflammation [12]. Impaired endothelial function is strongly associated with the development and progression of cardiovascular diseases, including atherosclerosis, hypertension, and heart failure [12]. Dysfunctional endothelium leads to the reduced bioavailability of vasodilators, increased expression of adhesion molecules, and enhanced pro-inflammatory states, ultimately contributing to vascular injury and disease pathogenesis [12]. Endothelial dysfunction is evaluated non-invasively in a laboratory setting using the flow-mediated dilation (FMD) technique, which is considered the gold standard for assessing endothelial function in humans [13,14]. This method measures the dilation of arteries in response to increased blood flow and shear stress, primarily mediated by nitric oxide (NO) [13]. FMD is typically measured in the brachial artery using high-resolution B-mode ultrasound, following a brief period of forearm ischemia induced by the placement and inflation of a cuff on the upper limb [15].

Obesity has been closely linked to endothelial dysfunction in humans. Notably, there is a positive correlation between BMI, endothelial dysfunction, and cardiovascular risk in young adults under 60, while this relationship weakens in older adults [16]. However, it remains unclear whether obesity-related endothelial dysfunction precedes the onset of cardiovascular disease or whether cardiovascular disease itself promotes endothelial dysfunction [16]. As previously mentioned, obesity is a pro-inflammatory state where adipose tissue secretes pro-inflammatory cytokines. Inflammatory cytokines, such as IL-1 and TNF-α, promote oxidative stress and reduce the phosphorylation of endothelial nitric oxide synthase within endothelial cells, leading to endothelial dysfunction [17]. Interestingly, endothelial cells from the visceral white adipose tissue of obese individuals, which is strongly associated with cardiovascular risk, have been found to express more inflammatory genes compared with endothelial cells from the subcutaneous white adipose tissue [18].

Another key mediator linking obesity with endothelial dysfunction is adiponectin. This hormone, produced by adipose tissue, regulates glucose homeostasis by inhibiting hepatic gluconeogenesis and enhancing insulin sensitivity [19]. Adiponectin also promotes endothelial integrity and function by increasing nitric oxide (NO) production and decreasing TNF-α activity within the endothelium, thereby reducing endothelial cell apoptosis. These effects are mediated through the activation of 5′ adenosine monophosphate-activated protein kinase (AMPK) and cyclooxygenase-2 (COX-2) [20]. Obese individuals exhibit decreased adiponectin levels, which are suggested to contribute to the development of endothelial dysfunction. Furthermore, Tsuchida et al. demonstrated that obese mice exhibited reduced adiponectin receptors in adipose tissue and skeletal muscle, leading to resistance to adiponectin’s actions [21].

Beyond obesity, scientific evidence links a high-fat diet (HFD) to endothelial dysfunction. Dow et al. found that healthy middle-aged adults consuming an HFD exhibited impaired endothelial function, as assessed by forearm blood flow responses to acetylcholine, compared with age-matched individuals on low-fat diets, regardless of body weight [22]. Similarly, Lambert et al. showed that young adults (mean age 23 ± 1 years) that consumed a high-fat diet had impaired endothelial function compared with those with healthier eating habits [23]. The underlying mechanisms for this association remain unclear. Recently, Jiang et al. demonstrated that a high-fat diet in mice led to an increased expression of nicotinamide adenine dinucleotide phosphate oxidase 2 (NOX2) and the receptor for advanced glycation end products (RAGE) in the ophthalmic artery, which was associated with elevated oxidative stress and endothelial dysfunction at the same site. The authors suggested that NOX2 and RAGE are involved in diet-induced autonomic dysfunction [24]. Additionally, García-Prieto et al. showed that HFD-induced endothelial dysfunction in mice is linked to dysfunctional endothelial AMPK, whose activation is known to increase nitric oxide (NO) availability within the endothelium [25].

## 4. Impact of Obesity and HFD on Autonomic Nervous System Dysfunction

At the same time, obesity has been linked to autonomic nervous system (ANS) dysfunction through various mechanisms. The activation of vagal afferent nerves, a key component of the ANS, following feeding due to gastric distension plays a significant role in satiety by transmitting food consumption information to the CNS [26]. Obese patients with insulin resistance exhibit increased sympathetic nervous system activity, which is known to affect the endothelium and potentially lead to cardiovascular events. The obesity-related increase in sympathetic activity is mediated by insulin action within the central nervous system. Higher insulin concentrations stimulate the hypothalamus to enhance glucose metabolism and activate sympathetic centers, resulting in increased sympathetic activity [26,27]. This leads to elevated arterial blood pressure, increased heart rate, and decreased heart rate variability.

Another mediator implicated in the connection between obesity and ANS dysfunction is leptin [26]. Leptin is a satiety hormone primarily produced by adipose tissue, providing energy-state information from the periphery to the CNS through its action on the arcuate nucleus (ARC) of the hypothalamus. Obese patients with excessive body fat mass have higher leptin plasma concentrations compared with lean individuals [26]. Leptin has been shown to activate the sympathetic nervous system through its action on the ARC. Animal studies have demonstrated that leptin administration in non-obese states leads to increased heart rate and arterial blood pressure, further supporting leptin’s role in sympathetic nervous system activation [28].

Other reported mechanisms involved in the association between obesity and ANS dysfunction include adrenergic receptor deficits in obesity and ANS activation related to obstructive sleep apnea syndrome [29,30]. The obesity-mediated ANS activation is reversible, as weight loss leads to decreased sympathetic nervous system activity, reduced blood pressure, and increased heart rate variability [31].

Autonomic nervous system (ANS) activity is influenced by diet composition. According to the literature, carbohydrate ingestion increases sympathetic activity and decreases parasympathetic activity. As for fat ingestion, the data are conflicting, showing either no change in the ANS or an increase in sympathetic activity [32]. Animal experiments have demonstrated that short periods of a high-fat diet (HFD) lead to increased sympathetic activity, heart rate, and arterial blood pressure, though these changes were accompanied with concomitant weight gain [33,34]. However, scientific evidence on the long-term effects of unhealthy diet patterns on the ANS is scarce. Interestingly, one human study involving healthy post-menopausal women found that vegetarians exhibited higher heart rate variability compared with omnivores [35]. The role of long-term diet composition on ANS dysfunction, regardless of body weight changes, needs to be better elucidated.

## 5. Potential Molecular Mechanisms Connecting Hypothalamic Inflammation with ANS Dysfunction and Endothelial Disorders

Systemic inflammation has been proven to impair endothelial function in humans. Hingorani et al. demonstrated that acute systemic inflammation induced by Salmonella typhi vaccination in healthy men significantly reduced the FMD of the brachial artery, indicating endothelial dysfunction [36]. Furthermore, Lind et al. investigated the effects of endotoxin-induced and vaccine-induced systemic inflammation in healthy men and observed similar impairments in endothelium-dependent vasodilation, as measured by local infusions of acetylcholine and forearm blood flow responses [37]. Additionally, scientific studies have demonstrated a link between hypothalamic inflammation, autonomic nervous system (ANS) dysfunction, and consequent endothelial disorders. These animal-derived data suggest that neuroinflammation of the hypothalamic paraventricular nucleus (PVN) and the subfornical organ (SFO) is primarily associated with autonomic dysfunction [38]. The SFO, a circumventricular organ closely associated with the hypothalamus, is crucial for regulating energy homeostasis, cardiovascular function, and osmoregulation through the control of hormones like angiotensin. Situated below the fornix near the foramina of Monro, the SFO contains neurons that project to various central nervous system (CNS) sites, including the PVN [39]. Additionally, the SFO lacks blood–brain barrier (BBB) protection. This allows it to integrate and process a wide range of peripheral hormonal and fluid balance information, along with inflammatory molecules. It then transmits this information to the PVN to trigger appropriate physiological responses [40]. Upon appropriate stimuli, decreased inhibition of nitric oxide and gamma-aminobutyric acid within the PVN leads to sympathetic hyperactivity. This is mediated by increased levels of angiotensin II, glutamate, and other neurotransmitters within the PVN [41]. The literature suggests several mechanisms that are likely involved in the connection between hypothalamic inflammation and ANS dysfunction.

Increased cytokines during neuroinflammation have been implicated in the activation of sympathetic flow. For example, elevated TNF-α within the PVN has been shown to increase sympathetic activity. The IKK/NF-κB signaling pathway is a major component of cellular regulation, playing a pivotal role in maintaining homeostasis. This pathway is activated by various stimuli, including pro-inflammatory cytokines, stress signals, and microbial infections. The IKK complex, composed of IKKα, IKKβ, and the regulatory subunit NEMO, orchestrates the phosphorylation and subsequent degradation of IκB proteins, liberating NF-κB to translocate into the nucleus. Once in the nucleus, NF-κB binds to specific DNA sequences to regulate the transcription of genes involved in inflammation, immune responses, cell survival, and apoptosis [42]. Purkayastha et al. demonstrated that the intracerebral injection of TNF-α in rodents resulted in the activation of the proinflammatory nuclear factor κB (NF-κB) and its activator IκB kinase-β (IKK-β) within the mediobasal hypothalamus, leading to sympathetic activation and consequently increased blood pressure. This effect was more pronounced in the pro-opiomelanocortin (POMC) neurons of the hypothalamus and was reversible after sympathetic suppression [43]. Recently, Wei et al. showed that TNF-α-mediated sympathetic activation and the resulting rise in blood pressure occur through the activation of epidermal growth factor receptor (EGFR), which, in turn, activates extracellular signal-regulated kinases 1 and 2 (ERK1/2) in the SFO and PVN [44]. Furthermore, TNF-α receptor knockdown in the SFO of mice resulted in decreased sympathetic activity and improved heart function, suggesting a potential therapeutic role for targeting TNF-α [45]. In addition to TNF-α, IL-1 has been shown to induce autonomic dysfunction. Rats treated with IL-1β exhibited elevated levels of IL-1β, COX-2, TNF-α, and TNF-α within the PVN and SFO; increased inflammation in these brain regions; and ANS dysfunction with increased sympathetic activity and increased blood pressure [46].

Hypothalamic inflammation involves increased gliosis, a reactive proliferation of glial cells within the CNS. Glial cells release gliotransmitters and regulate neuronal functions and synaptic activity. The hypothalamic PVN and SFO possess a distinct neuro-glial environment in which astrocytes inhibit neuron-to-neuron communication and neurotransmission. Under conditions of heightened hypothalamic activity, this neuro-glial environment undergoes morphological changes that enhance neurotransmitter diffusion to help the CNS achieve homeostasis [47]. Scientific evidence suggests that microglial dysregulation may play a role in hypothalamic inflammation-related autonomic dysfunction. Liu et al. demonstrated that rats with heart failure exhibit increased sympathetic activity, elevated cerebrospinal fluid concentrations of norepinephrine, and increased microglial inflammation, with a higher proportion of microglia polarized to the M2 phenotype within the PVN [48].

Angiotensin II is a key component of the renin–angiotensin system and well-known for its role in regulating blood pressure and fluid balance. However, recent findings highlight its broader functions as a pro-inflammatory and pro-fibrotic molecule, contributing to tissue damage and organ dysfunction in various diseases. By binding to its receptors, particularly the Angiotensin II type 1 receptor (AT1R), it promotes oxidative stress, mitochondrial dysfunction, and inflammatory responses. These mechanisms not only exacerbate disease progression but also suggest potential strategies for therapeutic interventions targeting Angiotensin II signaling in conditions such as inflammation, aging-related tissue injury, and autoimmune diseases [49]. Recently, Liu et al. also showed that angiotensin II activates microglia within the PVN, leading to the release of proinflammatory cytokines and the activation of the nuclear factor-kappa B (NF-κB) pathway, resulting in sympathetic overactivity and hypertension in male rats. Interestingly, the same study revealed that the administration of minocycline and pyrrolidine, which have anti-inflammatory and anticholinergic effects, respectively, reduced the microglial activation within the hypothalamus and consequently lowered the blood pressure [50].

Functional extracellular mitochondria, known as circulating mitochondria, have recently been recognized as novel inter-organ communicators. Recent studies have shown that obesity and a high-fat diet (HFD) cause mitochondrial dysfunction in white adipocytes, leading to the release of damaged mitochondria into the periphery. These damaged mitochondria then enter other cells through a process called intercellular mitochondria transfer [51,52]. Circulating mitochondria enter the cytoplasm of target cells and activate signaling pathways, such as cyclic GMP–AMP synthase (cGAS), which induces inflammation [53]. Abnormal circulating mitochondria may contribute to neuroinflammation-induced sympathetic overactivity. Zhang et al. demonstrated that mice with heart failure had damaged circulating mitochondria (with impaired respiratory activities) that stimulated cGAS activation within the SFO. This activation resulted in neuroinflammation characterized by microglial and astroglial activation, increased proinflammatory cytokines, and elevated production of endothelial ROS within the SFO, leading to sympathetic hyperactivity, as measured by renal sympathetic nerve activity, heart rate variability, and plasma norepinephrine concentration [54]. The same research team recently discovered that circulating mitochondria carrying cGAS act on the SFO endothelial phospholipase A2, which, in turn, induces neuroinflammation by activating the astroglial/microglial Integrin-alphavbeta3 within the SFO, resulting in central sympathetic hyperactivity. The authors concluded that cGAS is a potential therapeutic target for sympathetic hyperactivity in cardiovascular diseases [55]. Interestingly, sodium/glucose co-transporter 2 (SGLT2) has recently been found to inhibit SFO cGAS degradation, promoting neuroinflammation and sympathetic activity through the mechanisms described above [56]. This suggests the potential central role of widely used SGLT-2 inhibitors in ameliorating heart failure and other cardiovascular diseases.

As previously mentioned, neuroinflammation-related microglia activation is mediated by the activation of the Toll-like receptor 4 (TLR4) pathway, which stimulates the production of inflammatory cytokines. TLR4 itself has been linked to autonomic dysfunction: In rats, the activation of TLR4 (induced by injection of LPS, a specific ligand for TLR4) resulted in autonomic dysfunction, characterized by decreased heart rate variability and increased plasma norepinephrine release. Furthermore, TLR4 activation led to increased TNF-α levels and heightened microglia activation within the PVN, indicating that TLR4 is a potent mediator of neuroinflammation-induced sympathetic hyperactivity [57]. The inhibition of TLR4 in rats resulted in decreased levels of inflammatory cytokines TNF-α and IL-1β, as well as reduced nitric oxide synthase within the PVN, accompanied by decreased sympathetic activity, as suggested by reduced plasma norepinephrine levels and lower arterial blood pressure [58]. Additionally, Yu et al. demonstrated that activation of Angiotensin II type 1a receptors in the subfornical organ of rats led to the increased expression of inflammatory mediators, heightened gliosis within the PVN, and elevated sympathetic activity. The authors suggested that Angiotensin II could be a significant factor linking hypothalamic inflammation, sympathetic hyperactivity, and heart failure [59]. Interestingly, Biancardi et al. showed that Angiotensin II-mediated PVN inflammation and increased sympathetic activity were TLR4-dependent. Specifically, in rodents, Angiotensin II triggered microglia activation and increased ROS production within the PVN, and these effects were diminished in the absence of functional TLR4 [60].

From the above, it can be seen that hypothalamic inflammation within the PVN and SFO is strongly associated with autonomic dysfunction, particularly sympathetic hyperactivity. The implicated mechanisms include locally acting cytokines; gliosis activation; abnormal circulating mitochondria through stimulated cGAS activation, leading to microglial and astroglial activation; and finally, the activation of the TLR4 pathway, likely mediated by angiotensin II.

## 6. Discussion

In this narrative review, we explored the potential role of hypothalamic inflammation in linking obesity/HFD with autonomic dysfunction and subsequent endothelial disorders. Our literature research suggests that obesity and HFD-induced hypothalamic inflammation are associated with sympathetic hyperactivity. Key mechanisms include the action of inflammatory cytokines, such as TNF-α, which activates the NF-κB/IKK-β pathway, angiotensin II-activated microglia that further stimulate the NF-κB pathway, the action of circulating mitochondria on cGAS activation within the SFO promoting microglial activation, and the activation of the TLR4 pathway [43,44,48,50,54,55,57,58,59] (pp. 6–7). These molecular pathways of PVN and/or SFO inflammation lead to autonomic dysfunction, primarily through sympathetic hyperactivity, and result in endothelium-related disorders, such as arterial hypertension. Figure 1 illustrates the molecular mechanisms linking obesity/an HFD to neuroinflammation, while Figure 2 illustrates the potential pathways connecting neuroinflammation, sympathetic hyperactivity, and endothelial dysfunction.

It is important to note that the positive association between obesity/an HFD and hypothalamic inflammation has been proven in humans [3] (p. 2). Similarly, the relationship between obesity and endothelial dysfunction, as well as the link between obesity and endothelial dysfunction disorders, like arterial hypertension in humans, is well-established [61]. Regarding the association between obesity and sympathetic hyperactivity: although increased sympathetic activity is a major factor connecting obesity to cardiovascular disorders, measuring sympathetic tone in obese individuals has produced inconsistent results, likely due to the diverse methodologies used [62]. With the development of more accurate methods, it is now known that obesity is associated with increased sympathetic activity to muscles and kidneys, but not to the heart [62]. While clinical human studies have confirmed these associations, our understanding of the molecular mechanisms involved primarily comes from animal studies. More human studies are needed to better elucidate these mechanisms in order to develop effective prevention and treatment strategies for endothelial disorders associated with obesity–hypothalamic inflammation and autonomic dysfunction.

One promising molecule with the potential to enhance endothelial function by reducing neuroinflammation is the Glucagon-like peptide-1 (GLP-1) receptor agonist. GLP-1 agonists are a class of medications that mimic the action of the endogenous incretin hormone GLP-1 [63]. These drugs were first approved in 2005 for the treatment of type 2 diabetes. Due to their effect on body weight reduction, these drugs are now approved for the treatment of obesity, representing a revolution in obesity management [63]. Several studies have shown that GLP-1 agonists decrease arterial hypertension and reduce the risk of major adverse cardiovascular events in humans [63]. Furthermore, GLP-1 agonist administration in humans has been shown to decrease endothelial dysfunction and increase flow-mediated dilation (FMD) [64,65]. Animal studies have demonstrated that these drugs can reduce the sympathetic response [66]. The exact mechanisms through which GLP-1 agonists act on the endothelium and sympathetic flow remain unclear. GLP-1 agonists have been shown to exert anti-inflammatory effects [67]. Research indicates that GLP-1 agonists bind to GLP-1 receptors on immune cells, reducing the production and release of proinflammatory cytokines, such as TNF-α and IL-6 [67]. GLP-1 agonists can cross the blood–brain barrier, and GLP-1 receptors have been identified in various CNS regions, including the lateral hypothalamus, where they are crucial for the positive effects on body weight [68]. In vitro studies have shown that GLP-1 agonists decrease neuroinflammation by altering microglial function [69]. In mice, the activation of GLP-1 receptors within the CNS inhibited TLR-induced inflammation [70]. Thus, GLP-1 agonists can decrease neuroinflammation by suppressing microglia activation and the TLR pathway, both of which are associated with ANS and endothelial dysfunctions [48,57] (pp. 6–7). The administration of GLP-1 attenuates obesity, sympathetic hyperactivity, endothelial dysfunction, and cardiovascular disease [63,64,65,66]. GLP-1’s hypothalamic anti-inflammatory effect could be the link accounting for its positive effects on sympathetic flow and the endothelium, as repeatedly confirmed in clinical studies. More research in humans is essential to better understand the beneficial effects of GLP-1 agonists on neuroinflammation and to further elucidate the mechanisms that drive their positive impact on ANS function and endothelial health.

## Figures and Tables

**Figure 1 ijms-26-04047-f001:**
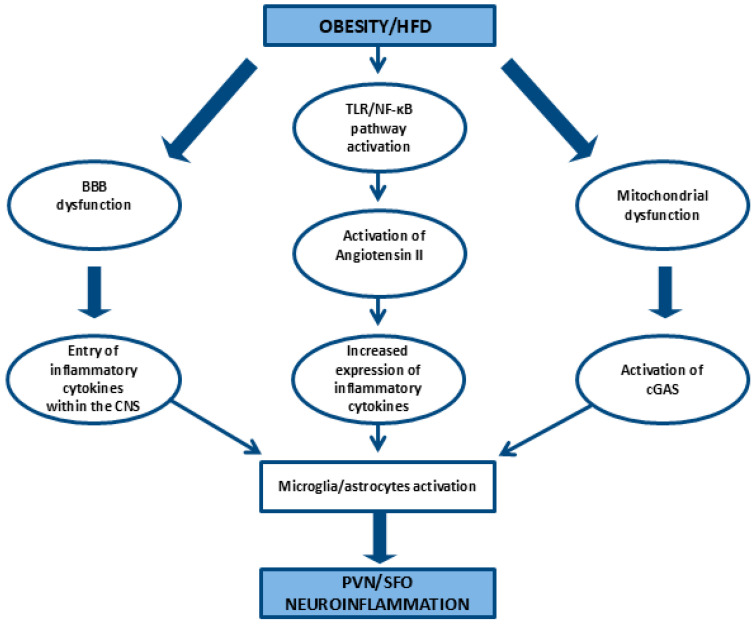
Molecular mechanisms linking obesity and high-fat diets to neuroinflammation. HFD: high-fat diet, BBB: blood–brain barrier, TLR: Toll-like receptor, NF-κb: nuclear factor-kappa B, CNS: central nervous system, cGAS: cyclic GMP–AMP synthase, PVN: paraventricular nucleus, SFO: subfornical organ.

**Figure 2 ijms-26-04047-f002:**
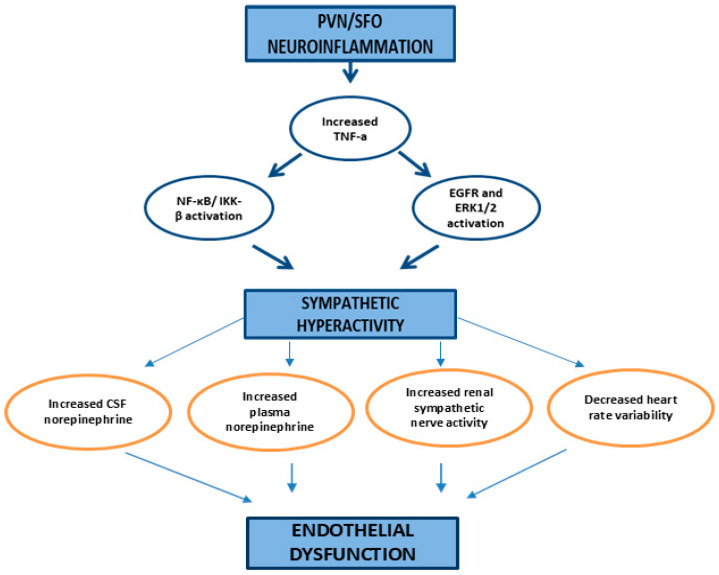
Hypothalamic inflammation-induced sympathetic hyperactivity and endothelial dysfunction pathways. TNF-a: tumor necrosis factor a, NF-κb: nuclear factor-kappa B, IKK-β: IκB kinase-β, EGFR: epidermal growth factor receptor, ERK1/2: extracellular signal-regulated kinases 1 and 2, CSF: cerebrospinal fluid.

## Data Availability

No new data were created or analyzed in this study. Data sharing is not applicable to this article.

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
