# Peer review of "Obesity- and High-Fat-Diet-Induced Neuroinflammation: Implications for Autonomic Nervous System Dysfunction and Endothelial Disorders"

_ijms, 2025, doi:10.3390/ijms26094047_

Round 1

Reviewer 1 Report

Comments and Suggestions for Authors

The present narrative article discusses how obesity-induced neuroinflammation contributes to autonomic nervous system dysfunction and endothelial dysfunction. The article is well-structured and clearly written. However, there are a few suggestions that could be incorporated into the revised version to enhance clarity and improve the overall understanding for readers.

Scientific suggestions

  1. The authors could consider adding a figure to illustrate how these mechanisms are interlinked, particularly highlighting the major contributing factors.
  2. As the authors mention in the abstract and discussion sections, the NF-κB/IKK-β pathway and angiotensin II play major roles. However, these components are not discussed in detail. It would be beneficial to include a dedicated sub-section addressing the relevant signaling pathways to enhance clarity and understanding for the readers.
  3. Although this is a narrative review, it would be beneficial to include some case studies that provide a mechanistic overview of inflammation and endothelial dysfunction.
  4. Figure 1 does not clearly illustrate the key mechanisms. It should be revised to clearly depict how obesity and HFD contribute to endothelial dysfunction.

Author Response

Reviewer #1: The present narrative article discusses how obesity-induced neuroinflammation contributes to autonomic nervous system dysfunction and endothelial dysfunction. The article is well-structured and clearly written. However, there are a few suggestions that could be incorporated into the revised version to enhance clarity and improve the overall understanding for readers.

Reviewer #1, Comment No 1: The authors could consider adding a figure to illustrate how these mechanisms are interlinked, particularly highlighting the major contributing factors.

Answer to Comment No 1:  We are grateful for the positive comments and the valuable suggestion of Reviewer #1. In response to this comment, we have replaced Figure 1 with two new figures in the manuscript (Discussion section, Lines 339-368). These updated figures illustrate the main probable molecular factors implicated in the association between obesity/high-fat diet, hypothalamic inflammation, autonomic nervous system dysfunction and endothelial disorders.

Reviewer #1, Comment No 2: As the authors mention in the abstract and discussion sections, the NF-κB/IKK-β pathway and angiotensin II play major roles. However, these components are not discussed in detail. It would be beneficial to include a dedicated sub-section addressing the relevant signaling pathways to enhance clarity and understanding for the readers.

Answer to Comment No 2:  We thank the Reviewer for this valuable suggestion. We fully agree that these signaling pathways require a more detailed discussion in the manuscript. In response, we have added the following paragraphs to the revised version (Section 5 Lines 228-235 and Lines 263-271) along with two new references (Ref 42 and 49):

’The IKK/NF-κB signaling pathway is a major component of cellular regulation, playing a pivotal role in maintaining homeostasis. This pathway is activated by various stimuli, including pro-inflammatory cytokines, stress signals, and microbial infections. The IKK complex, composed of IKKα, IKKβ, and the regulatory subunit NEMO, orchestrates the phosphorylation and subsequent degradation of IκB proteins, liberating NF-κB to translocate into the nucleus. Once in the nucleus, NF-κB binds to specific DNA sequences to regulate the transcription of genes involved in inflammation, immune responses, cell survival, and apoptosis.’’

’Angiotensin II is a key component of the renin-angiotensin system and well-known for its role in regulating blood pressure and fluid balance. However, recent findings highlight its broader functions as a pro-inflammatory and pro-fibrotic molecule, contributing to tissue damage and organ dysfunction in various diseases. By binding to its receptors, particularly the Angiotensin II type 1 receptor (AT1R), it promotes oxidative stress, mitochondrial dysfunction, and inflammatory responses. These mechanisms not only exacerbate disease progression but also suggest potential strategies for therapeutic interventions targeting Angiotensin II signaling in conditions such as inflammation, aging-related tissue injury, and autoimmune diseases.’’

Kindly note that the original title of Section 5, 'Hypothalamic Inflammation, ANS Dysfunction, and Endothelial Disorders,' has been replaced to 'Potential Molecular Mechanisms Connecting Hypothalamic Inflammation with ANS Dysfunction and Endothelial Disorders' to more accurately reflect the section's objective.

Reviewer #1, Comment No 3: Although this is a narrative review, it would be beneficial to include some case studies that provide a mechanistic overview of inflammation and endothelial dysfunction.

Answer to Comment No 3:  We thank the Reviewer for this insightful suggestion, which improves the quality of our manuscript. In response, we have incorporated a new paragraph and two references highlighting human case studies that demonstrate the link between systemic inflammation and endothelial dysfunction. (Section 5, Lines 202-208, Ref 36,37)

’Systemic inflammation has been proven to impair endothelial function in humans. Hingorani et al. demonstrated that acute systemic inflammation induced by Salmonella typhi vaccination in healthy men significantly reduced FMD of the brachial artery, indicating endothelial dysfunction. Furthermore, Lind et al. investigated the effects of endotoxin-induced and vaccine-induced systemic inflammation in healthy men and observed similar impairments in endothelium-dependent vasodilation, as measured by local infusions of acetylcholine and forearm blood flow responses.’’

Reviewer #1, Comment No 4: Figure 1 does not clearly illustrate the key mechanisms. It should be revised to clearly depict how obesity and HFD contribute to endothelial dysfunction.

Answer to Comment No 4: We thank the Reviewer for the comment regarding Figure 1. In response, we have replaced the original figure with two more detailed and inclusive figures. We hope that the new figures effectively address the concerns raised. (This also addresses comment No. 1).

Reviewer 2 Report

Comments and Suggestions for Authors

This manuscript studies the implications of obesity and high-fat diet-induced neuroinflammation on autonomic nervous system dysfunction and endothelial disorders. It is a comprehensive review, but two or more schemas must be included to illustrate how these systems affect each other. The figures would help readers follow and understand.

Author Response

Comment of Reviewer #2:  This manuscript studies the implications of obesity and high-fat diet-induced neuroinflammation on autonomic nervous system dysfunction and endothelial disorders. It is a comprehensive review, but two or more schemas must be included to illustrate how these systems affect each other. The figures would help readers follow and understand.

Answer to Reviewer No 2: We sincerely thank Reviewer #2 for their thoughtful comment and valuable suggestion. To enhance the clarity and comprehensiveness of our manuscript, we have replaced the original Figure 1 with two new, detailed figures (Discussion section, Lines 339-368). Figure 1 illustrates the molecular mechanisms linking obesity/high-fat diet to neuroinflammation, while Figure 2 depicts the potential pathways connecting hypothalamic inflammation, sympathetic hyperactivity, and endothelial dysfunction. We hope these updated figures address the Reviewer's concerns and improve the overall understanding for readers.
